# Biomedical and Antioxidant Potentialities in Chilli: Perspectives and Way Forward

**DOI:** 10.3390/molecules27196380

**Published:** 2022-09-27

**Authors:** Solanki Bal, Amit Baran Sharangi, Tarun Kumar Upadhyay, Fahad Khan, Pratibha Pandey, Samra Siddiqui, Mohd Saeed, Hae-Jeung Lee, Dharmendra K. Yadav

**Affiliations:** 1Department of Vegetable Science, BCKV-Agricultural University, Mohanpur 741252, India; 2Department of Plantation, Spices, Medicinal & Aromatic Crops, BCKV-Agricultural University, Mohanpur 741252, India; 3Department of Biotechnology, Parul Institute of Applied Sciences and Centre of Research for Development, Parul University, Vadodara 391760, India; 4Department of Biotechnology, Noida Institute of Engineering & Technology, Greater Noida 201306, India; 5Department of Biotechnology, School of Engineering and Technology (SET), Sharda University, Greater Noida 201310, India; 6Department Health Services Management, College of Public Health and Health Informatics, University of Hail, Hail P.O. Box 2240, Saudi Arabia; 7Department of Biology, College of Sciences, University of Hail, Hail P.O. Box 2240, Saudi Arabia; 8Department of Food & Nutrition, College of Bionano Technology, Gachon University, 1342 Seongnamdaero, Seongnam-si 13120, Korea; 9Department of Pharmacy, Gachon Institute of Pharmaceutical Science, College of Pharmacy, Hambakmoeiro 191, Gachon University, Incheon 21924, Korea

**Keywords:** *Capsicum annuum*, capsaicin, capsaicinoids, TRPV1 receptor, antioxidants, anticancer properties, health benefits

## Abstract

Worldwide, since ages and nowadays, traditional medicine is well known, owing to its biodiversity, which immensely contributed to the advancement and development of complementary and alternative medicines. There is a wide range of spices, herbs, and trees known for their medicinal uses. Chilli peppers, a vegetable cum spice crop, are bestowed with natural bioactive compounds, flavonoids, capsaicinoids, phytochemicals, phytonutrients, and pharmacologically active compounds with potential health benefits. Such compounds manifest their functionality over solo-treatment by operating in synergy and consortium. Co-action of these compounds and nutrients make them potentially effective against coagulation, obesity, diabetes, inflammation, dreadful diseases, such as cancer, and microbial diseases, alongside having good anti-oxidants with scavenging ability to free radicals and oxygen. In recent times, capsaicinoids especially capsaicin can ameliorate important viral diseases, such as SARS-CoV-2. In addition, capsaicin provides an ability to chilli peppers to ramify as topical agents in pain-relief and also benefitting man as a potential effective anesthetic agent. Such phytochemicals involved not only make them useful and a much economical substitute to wonder/artificial drugs but can be exploited as obscene drugs for the production of novel stuffs. The responsibility of the TRPV1 receptor in association with capsaicin in mitigating chronic diseases has also been justified in this study. Nonetheless, medicinal studies pertaining to consumption of chilli peppers are limited and demand confirmation of the findings from animal studies. In this artifact, an effort has been made to address in an accessible format the nutritional and biomedical perspectives of chilli pepper, which could precisely upgrade and enrich our pharmaceutical industries towards human well-being.

## 1. Introduction

Chilli pepper, botanically *Capsicum annuum* L., is an indispensable vegetable cum spice crop grown commercially worldwide for its immature green and red ripe fruits. Universally, the crop is consumed and appreciated for its flavor, colour, aroma, texture, and preserving foods. The crop was probably first used among folks as medicinal plants for its rich bountiful and diversified nutrients, even before it was used in cooking [1]. The understanding on use of such plants helped the masses to sustain since ancient times [2]. As medicinal plants, chilli peppers are known to possess many biochemically and pharmacologically active compounds, but the health benefits pertaining to chilli consumption are in question among health specialists, whether it can be single-handedly or in synergy modulated to improve human health [3]. Chilli fruit are known to possess up to 15 or more capsaicinoids, a preponderance of which are dihydrocapsaicin, capsaicin, nordi-hydrocapsaicin; 23 flavonoids including apigenin, luteolin, quercetin [4,5,6,7,8], and some catechins and cyanidins are notable. Thus, by consuming chilli one not only ingests vitamins and nutrients but capsaicinoids and flavonoids as well. On consuming capsaicin in chilli stimulates gastric secretions and stimulation along with stronger neural networks; in addition, capsaicin is associated with antioxidant action and can bind cancer cells and destroy them [9]. Besides, role of capsaicin in chilli having direct carcinogenic effect and association on consuming spicy chilli peppers and mortality rate in population have not yet been demonstrated. In past decades, articles were published by researchers based mainly on beneficial and adverse effects of capsaicinoids and flavonoids; gastronomic aspects; functional and nutraceutical properties of different bioactive compounds; phytochemical and pharmacologically active compounds of chilli upon human health, but experimental results are debatable and require human trials based on biological models. Thus, health benefits on chilli consumption need to be further investigated using theoretical and experimental models as well as biological models. In this artifact, an effort has been made to broaden the horizon of numerous and divergent potentialities of chilli pepper and its benefit pertaining to human health.

## 2. Chilli: A Brief Account

Chillies are one of the well-known crops of family Solanaceae belonging to the genus *Capsicum.* There are 5 domesticated chilli species namely, *C. annuum*, *C. baccatum*, *C. chinense*, *C. pubescens* and *C. frustescens*. Among all these species, *Capsicum annuum* is economically important and widely cultivated *Capsicum* sp. The crop is native to east-central Mexican region and was first independently cultivated in different regions of America including Central Mexico, highlands of Bolivia and the Amazon [10]. Botanically fruits are berries with varied shapes among different species [11,12]. This herbaceous crop can grow tall up to 1 m. Structure of leaves are mostly lanceolate to ovate-lanceolate, ovate or oblong-ovate. Flowers are white or purple-tinged color and ideally very small in size. Chilli fruits are mostly green when immature but can be red or green at maturity. Fruits can grow at varied length ideally up to 15 cm; seeds are mostly reniform or discoid in shape, pale-yellow in color. The crop requires warm, humid climate to survive [12] and grows well under tropical climate.

## 3. Traditional Medicinal Uses of Chilli

Use of chilli especially in medicinal practices since pre-Hispanic times from Aztecs and Mayans were well documented in several books and manuscripts. The most significant among them is *Libellus de Medicinabilus Indorum Herbtis* written in 1522 by Martin de la Cruz [13]. Chilli is native to Mexico and Central America and indigenous Mayans of Mesoamerica reported 32 (approximately) different health related uses in chilli [14], which includes, bellyache, arthritis, dermatitis, rheumatism as well as attenuating snake bites. Nonetheless, use of chilli was not only constricted within the boundaries of Latin America, it rolled out to other parts of the world through cultivation and consumption for its medicinal values. In “Blue Beryll,” the Tibetan medical classic text, wider aspects of medicinal uses of chilli were mentioned, including medication of hemorrhoids, leprosy, and edema [15]. Fruits are known to apply topically in pain disorders particularly migraine, neuropathy, psoriasis, trigeminal neuralgia, cluster headache, and herpes zoster. In Africa, fruits are considered more useful as antitussive agents, pulmonary disinfectant and antispasmodic agents [16]. Furthermore, chilli fruits aid in treating atherosclerosis, muscle tension, flatulence, stroke, dyspepsia, and loss of appetite [17]. At present, on a daily basis, chilli fruits are consumed both in fresh and dried form by a wide section of the world population either as spices, additives, or supplements.

## 4. Nutritional Profile

### 4.1. Vitamins

Chillies are known to possess wide range of vitamins especially vitamin A (carotene and C (ascorbic acid), vitamins B_2_, vitamin B_3_, vitamin B_6_, vitamin B_9_ and vitamin E. Among different vitamins, vitamin C is the only vitamin with strong antioxidant properties that can scavenge free radicals. This happens due to its conjugation of enediol structure for the carbonyl group present in a lactone ring [12]. Such reducing power aids in preventing chronic and degenerative diseases. Pungency in chilli pepper has a directly correlation with ascorbic acid content. Besides, concentration of ascorbic acid increases as the fruit reaches its physiological maturity. Studies revealed that vitamin C has the potentiality in abundance in managing body weight. Vitamin C heals cellular damage, strengthens the immune system and prevents respiratory infections. In a trial among Egyptian chilli genotypes it was found that ascorbic acid content to be high in fresh chillies. Upon drying, chilli loses out most of its vitamin C. The reverse trend was observed for vitamin A upon drying, it increases by 100 times. Vitamin A serves as an anti-inflammatory agent, immunity booster and good retinoid activity. Researchers further revealed in their studies that the level of carotenes and ascorbic acid is six times higher than citrus fruits. In addition, the vitamin C content in red-fruited chillies is twice higher than those of the green fruited ones [18].

### 4.2. Nutritional Profile of Chilli Fruits across Species (per 100 g of Edible Portion)

From the below Table 1 it can be well understood that wide array of chilli species provides ample amount of conventional nutrients. Empirical works had indicated on consuming chilli in either fresh or dry form imparts several health benefits [19].

### 4.3. Phytonutrients and Phytochemical Profiles

In chilli, carotenoids that are yellow-orange-red lipophilic pigments exist in numerous structures, forms at different maturity stages. In chilli fruits, at green colored stage in the chloroplast, an amalgamation of carotenoid and chlorophyll is found. At the intermediary stage of maturity, wide variety of compounds namely, α-carotene, β-carotene, β-cryptoxanthin, zeaxanthin and lutein are seen to be present. De novo, animals aren’t able to fathom carotenoids but can obtain them through daily diets. Carotenoids have the ability to protect bodily tissues from oxidative damages. At red-ripe stage, this carotenoid pigment transforms into capsanthin (6.97 mg/100 g DW), capsorubin (4.50 mg/100 g DW), and 5,6-epoxide (0.79 mg/100 g DW), which produces red colored pigment [22]. In research by Hassan et al. [22], it was revealed that the red color of chilli fruits are induced by a unique group of keto-carotenoids (which includes capsorubin, capsanthin, and cryptocapsin) having κ-ring as end groups. Carotenoids with such κ-ring as end groups have high potential in scavenging oxidative free radicals [23]. Thus, this red color of hot pepper is caused due to carotenoids and is contemplated as one of the ideal sources of β-carotene (28.47 mg/100 g FW), as reviewed by Arimboor et al. [24]. The concentration of phenolic acids, flavonoids in chilli peppers increases as the fruit approaches maturity. In a study by Nagy et al. [25], flavonoids (29.9~37.4 µg/g DW), catechin (6.6~12.3 µg/g DW), derivatives of vanillic acid and narigenin-diglucoside are the most commanding polyphenols in hot peppers. Jeong et al. [26] revealed that flavonoid activity increased at a constant level of ascorbic acid and caffeic acid and has the ability minimize oxidative damage. Red pepper seed and pericarp extracts serves as potent antioxidant agent. Fruit pericarp of chilli pepper has a strong total phenolic (TPC) and flavonoid (TFC) content [27] that scavenges free radicals and can form chelates with ferrous compounds. Besides, in seeds, high scavenging strength is due to ABTS [2,2’-azino-bis(3-ethylbenzothiazoline-6-sulfonic acid)], which works best against superoxide anion radical.

Phytochemicals in chilli peppers especially phenols, quercetin, luteolin and capsaicinoids [28,29,30] serves protection against oxidative stress, diabetes, cancer insurgence, neurological disorders such as Alzheimer’s and Parkinson’s disease [28,31,32]. In a study by researchers on various non-pungent varieties/cultivars ‘California Wonder’ (red pepper), ‘Lamuyo’ (yellow pepper) and ‘Italian Sweet’ (green pepper) exhibited wide span of phenolic classes, which includes 3,7-di-*O*-α-L-rhamnopyranoside, narigenin-7-*O*-β-D-(3″-p-coumaroyl)-glucopyranoside and quercetin [33]. In non-pungent chilli pepper genotypes, dihydrocapsiate, capsiate and nordi-hydrocapsiate [34] are present. The composition of phytochemicals varies with genetics, developmental stage and environmental conditions of chilli pepper [35]. Wide array of quercetin are observed in *C. annuum*, which includes four quercetin glycosides viz. 3-*O*-rhamnoside-7-*O*-glucoside and quercetin 3-*O*-rhamnoside. The quercetin, 3-*O*-glucoside-7-*O*-rhamnoside along with rhamnoside-glucoside is bonded at C-3 or C-7 position. Besides, five luteolins had been identified on the pericarp of chilli pepper fruits especially luteolin 6-*C*-pentoside-8-*C*-hexoside, luteolin 6-*C*-hexoside-8-C-pentoside, luteolin 6-*C*-hexoside, luteolin 8-*C*-hexoside, luteolin-*C*-glycosides, luteolin 6,8-di-*C*-hexoside along with two luteolin-*O*-glycosides, luteolin 6,8-di-*C*-hexoside, luteolin (apiosyl-acetyl)-glucoside and luteolin 7-*O*-(2-apiosyl)-glucoside [35]. Additionally, two apigenin C-glycosides namely apigenin 6,8-di-*C*-hexoside and apigenin 6-*C*-pentoside-8-*C*-hexoside as were also identified in chilli pepper fruits [36]. Furthermore, C-glycosides that were derived from *C. annuum* L., var. Capel Hot includes luteolin 6-C-glucoside, apigenin 6-*C*-glucoside-8-*C*-arabinoside and luteolin 6,8-di-*C*-glucoside [37].

### 4.4. Bioactive Compounds

Wide array of distinct bioactive compounds have been found to be present in different chilli species (Figure 1). These distinct bioactive compounds include capsaicin, homodi-hydrocapsaicin, nordi-hydrocapsaicin, homo-capsaicin and 6,7-dihydrocapsaicin [38]. Chilli pepper fruits vary in their shape, size, color, pungency and flavor and such variation falls out due to genotypic variation, fruit maturity, growing conditions and post-harvest maneuvers. Pungency in chilli depends upon the concentration of capsaicinoids especially capsaicin, which varies in the range of 600 to 13,000 ppm depending upon the genotypes. Capsaicin, the pungent principle in chilli is responsible for causing burning sensation and such irritation can last for hours after ingestion. Other bioactive compounds that are known to be present in chilli include flavonoids and carotenoids. These compounds are known to alleviate and maintain human immune system [39]. Certain principal compounds especially 2-isobutyl-3-methoxypyrazine, linalool, hexanal, trans-2-hexenal, 3-carene and 2, 3-butanedione are known as the constituents of commercial fresh chilli (*Capsicum annuum* L.) at different maturity stages of the crop [40]. Confirmation on existence of such compounds in chilli was made confirmed by analytical techniques, such as gas chromatography, HPLC, gas chromatography and mass spectrophotometry [18]. Among all these capsaicinoid compounds mentioned above, capsaicin, which is chemically 8-methyl-*N*-vanillyl-6-nonenamide occupies 80–90% of the total capsaicinoid group and is considered as ‘sine qua non’ factor of the genus *Capsicum*. The antimicrobial and antioxidant properties of such compounds have drawn interest of both academic and industrial research.

#### 4.4.1. Capsaicin and Its Medicinal Horizon

Capsaicin (Figure 2) content in chilli is governed by a major gene, but its mode of action is governed by polygenes acting in an aggregated manner determining various degrees of pungency. It is well absorbed in the body owing to its unique chemical structure [41,42] when consumed orally or topically, holding out to 94% absorption in the body. Capsaicin imparts several pharmacological properties benefitting humans by treating hypertension, ischemic heart disease, chronic pain syndrome, hypoglycemic effects. Besides, capsaicin possesses antimicrobial properties i.e., it can act as potent natural inhibitor against pathogenic micro-organisms in food [43]. Vanilloid receptor subtype 1, a receptor recognized to have the potential to bind capsaicin in the plasma membrane, where it forms a non-selective cation channel to bring in pleiotropic effects exerted mostly by capsaicin and its analogs [44]. Furthermore, capsaicin prevents cytotoxic effects by enabling its wide array of signaling mechanisms, including activation or up-regulation of p53 [45], activation of (STAT) or transcription family of proteins including the NF-kB pathway [46] and by generating Reactive Oxygen Species (ROS). Experimentally, overdosing of capsaicin has proven to be hazardous. However, the said compound does not result in any pernicious hazards to tissues. Studies revealed that Capsaicin is potent enough to have LD_50_ in mice, which is 47.2 mg/kg approximately [12]. Studies in experimental mice had further revealed that capsaicin had the potentiality in destroying cancer cells in prostate keeping normal cells unharmed. The mechanism engaged in desolation of such cancer cells aided by capsaicin is not known till date but the findings has gently ignited the hope that capsaicin can be used in developing potent anticancer therapeutic formulation [44]. Capsaicin concentration has been found considerably high and is effective against neo-plastic cells, but it has found quiet impossible to derive in-vivo from chillies especially in daily diet for humans. Capsaicin is known to have chemo-preventive effects by reducing its malignant growth particularly of lung-, colorectal-, gastric-, pancreatic-, breast- and prostate cancer [45]. It can reduce the size of tumor by decreasing the rate of progression of its growth. Till date, studies could not establish either of the facts that capsaicin could cure cancer entirely or destroy tumor completely. Hence, it requires further studies on this aspect to put in place the exact genetics pertaining to capsaicin-provoked destruction of cancer cells along with the genetic mechanisms it follows [47]. Besides, capsaicin has anti-inflammatory potency. Neuropeptide P is analogous to inflammation and capsaicin is reported to be potent inhibitor of substance P. Studies in animals revealed that capsaicin has the capability to delay arthritis and deplete inflammation. Furthermore, Capsaicin has been reported to serve as neuroprotective agent against diabetic neuropathy and pain associated with arthritis and psoriasis [48]. Studies on 200 humans had proved capsaicin to be a natural analgesic [49] and can further reduce oxidation of triglycerides in humans across gender [50]. The pungent principle has also proved to be potent cardio protective agent by reducing blood cholesterol, aggregation of platelets, triglyceride levels and also enhances the fibrin solubility [51].

In the Figure 2, from the substrate phenylalanine the enzymes Phenyl ammonia-lyase in short PAL, Cinnamate (CA) 4-hydroxylase in short C4H, 4-Coumarate-CoA-Ligase in short 4CL, Hydroxycinnamoyltransferase in short HCT, Catechol-*O*-Methyltransferase in short COMT, Hydroxycinnamoyl-CoA-hydratase-lyase in short HCHL, putative Aminotransferase in short pAMT through different compounds lead to the formation of Vanillylamine, whereas, from the substrate Pyruvate, the enzymes ALS (Acetolactate synthase), AHRI (Acetohydroxy acid reductoisomerase), DHAD (Dihydroxyacid dehydratase), BCKDH(Branched-chain α-ketoacid dehydrogenase complex), FAT (Fatty acid Transport), ACS(Acetyl-coenzyme A synthetase)through different compounds leading to the development of 8-methyl-6-nonenoyl-CoA. The compound Vanillylamine when converted to 8-methyl-6-nonenoyl-CoA via CS-AT (3), capsaicin results.

#### 4.4.2. Capsaicin and TRPV 1

Transient Receptor Potential Cation Channel Subfamily V Member 1 (TRPV1 aka) is a temperature-sensitive ion channel that is engaged in pain sensation, which acts a pivotal part in avoiding heat and pain [52]. TRVP1 is a protein in humans that is encoded by TRPV1 gene and is frequently found in the peripheral nervous system especially in nociceptive neuron. It is also popularly known as the vanilloid receptor 1 and capsaicin receptor [53]. TRPV1 serves as a cation channel (which is non-selective) especially for capsaicin and thereby channelizes the calcium (Ca^2+^), sodium (Na^+^) ions movement (Figure 3) especially into the sensory cells during the process of activation [54]. Capsaicin is found to be TRPV1 agonist. It triggers in activating mechanisms of TRPV1. Besides, capsaicin also activates TRPV1 causing a burning sensation and pain in human body. Simultaneously, such activation of TRPV1 also helps in regulating body temperature [55]. A study by Leung [56] validated that upon consuming diet rich in capsaicin content, expression mRNA present in TRPV1 receptors gets increased especially in mice adipose tissues along with an optimum reduction of 24% of visceral fat. Furthermore, capsaicin can trigger anti-cancer effect particularly in breast cancer on activation of TRPV1 receptors and thereby induces necrosis and apoptosis in the target cells. Besides, such capsaicin-provoked TRPV1 activation had also resulted in imparting a positive symbiotic effect with an anti-cancer drug named pirarubicin (THP) for treating bladder cancer [57,58].

#### 4.4.3. Capsaicinoids

Capsaicinoids (CAPS), typical molecular alkaloid structure observed only within the genus *Capsicum*. Such structure is comprised of vanillyl group bonded by an alkyl and amide chain [59]. This unique molecular structure provides CAPS an amphiphilic nature with no behavior of basicity, akin to alkaloids in general [60,61]. Compounds that constitute the class capsaicinoids are usually four in number including 2 major and 2 minor compounds. Such major compounds include capsaicin and dihydro-capsaicin; minor compounds include homocapsaicin and nordi-hydrocapsaicin. In red fruits of *Capsicum frutescens*, 15 capsaicinoids have been identified and characterized by [62], which includes dihydrocapsaicin, nordi-hydrocapsaicin and capsaicin with norcapsaicin, nordi-hydrocapsaicin, n-vanillyl-nonanamide-homocapsaicin I, homocapsaicin II, norcapsaicin, n-vanillyl-octanamide, homodihydrocapsaicinisomes I, and homodihydrocapsaicinisomes II, including n-anillyl-decanamide as a minor compound.

##### Bioactivities of Capsaicinoids

The unique structure of capsaicinoid (CAPS) when exposed to temperatures over 80 °C gets readily decomposed to 50% of its original concentration. Such breakdown is assigned to bifurcation of C-N bond resulting in production of vanillyl group and acyl chain [60,61,62]. CAPS possessing this vanillyl group within have been observed to behave as a proton donor. Such structure strengthens the molecules to stabilize radicals, thereupon, interacts with enzymes and cellular membranes [63,64]. This claims that the bio-activity of CAPS is assigned to the presence of vanillyl group in the structure [65,66]. Vanilloid-1, a potential transient receptor is accountable for neural response to nausea, pain, osmolarity and temperature variations generating sensation of heat and/or pain. Such exemplary leads body to adjust all possible dangerous situations (Figure 4). The receptor, VR1, comprised of cation channel (non-selective in nature) backed by proton sites, present within vanilloid functional groups and different molecules [66]. This VR1 has multi-dimensional effects, such as anti-inflammatory, anti-cancer, antitumoral and provides pain relief and gastrointestinal effects [59,66,67,68]. Due to amphiphilic nature of CAPS, VR1 remains attached to CAPS without suffering any cleavage and thus the biological activity of CAPS has a close bearing on desensitization of VR1 [65,66] (Figure 5). Such process of interplay of VR1 with CAPS was narrated by Hayman and Kam [65]. On solubility point of view, capsaicinoids are reported to be soluble in solvents having medium to low polarity such as methanol, acetonitrile, ethanol and alcohol. Among all these solvents CAPS has highest solubility in Acetonitrile and thus it is the most used solvent in CAPS extraction. Besides, the functional group that is phenolic in nature warrants CAPS to undergo transformation when exposed to alkaline conditions, where, it forms a phenolate ion. This, in turn, forms salt along with metals accelerating its solubility especially in aqueous solution [69].

##### Factors Governing CAPS Concentration

The genus *Capsicum* has a huge diversity due to the existence of large number of cultivars within each species. Thus, it becomes extremely hard to confirm or accomplish factors that influence CAPS concentration. Inspite of this perplexity, environmental-ecological conditions have a great impact in affecting CAPS profile [70,71]. CAPS are synthesized particularly in the placenta of chilli fruit. Genetically it is determined by the presence of *Pun1* or *pun1* gene. This ascertains that the production and accumulation of CAPSs has a close bearing in the expression of specific alleles in DNA. The concerned gene is linked with AT3- or EST- type cofactors. It induces quantitative effect of the gene and thereby imposes wide range of variations in the pungency of fruit. Not all chilli fruits are pungent, chilli varieties having recessive *pun 1* gene possesses sweet fruits. Considering the intrinsic factors there is no solid evidence on mechanism governing transcription of these genetic markers on complex metabolic network of CAPS [72,73,74,75]. Any minor modifications in its transcription can result in heterogeneous behavior pertaining to CAPS profile observed across different species and cultivars. However, genotypes are considered as the most important factor to affect concentration of CAPS [71,74,75,76,77,78,79]. An external factor that alters CAPS content includes soil fertility, water availability, temperature, and amount of sunlight [71,80].

In accordance to the cultivar type, sunlight and temperature had been set out to ignite negative and positive correlations although there is no homogeneity in the behavior of the species [77,81]. Such correlation between temperature and sunlight with CAPS content can be postulated with compensatory mechanism either of competition or protection process. A positive correlation depicts that CAPS can act as protective metabolites against oxidative stress. Whereas, negative correlation pops up when plants are exposed to higher temperatures, uses photosynthates during their growth period in order to reduce lignin-like compounds acting in a metabolic pathway competitively reducing CAPS content as a consequence [71,82]. The heterogeneity of these factors and the subsistence of cultivars that maintains a uniform CAPS content exhibits that the influence of such factors is insignificant for economical and phytogenetic approaches in CAPS profile [78].

Furthermore, soil fertility and its effect have less concordance within literatures. Earlier works have proved that either nitrogen or potassium had imparted a positive or negative associationship with CAPS content [83,84,85,86]. Besides, cognitive to temperature effects, whether the water holding capacity do present a heterogeneous correlation or decreases CAPS content are prudent in several literatures [87,88]. In previous studies the research was carried out without considering other factors, which imposes effect on fertility of soil, especially, micronutrients, microbial biomass (residual), granulometry and organic carbon and relief type [89]. There is thus a lacuna in research on how soil fertility affects CAPS concentration.

CAPS content varies with respect to ripening stage in order to provide simultaneously both protective as well as pollinating properties [59,90]. A slight variation is seen to occur when the fruit approaches maturity, which is usually after 40 to 50 days where a majority of CAPS content is produced by the cultivars. Such variation results due to decrease in peroxidase isoenzymes responsible for CAPS catabolism [91,92]. It has been reported that such shift in each of CAPS content varies with respect number of days after anthesis [93].

#### 4.4.4. Capsinoids

In the genus *Capsicum*, a set of substances called capsinoids within CAPS group are blended exclusively in pungent and non-pungent cultivars [94]. CAPS for decade had been widely utilized by industries due to their pungency and bioactive properties. Although their applications in human are limited due to high pungency and toxicological effect produced in humans [66,95,96]. Thus, a promising substitute capsinoids has popped up. The chemical structure of capsinoids is analogous to that of CAPS and is of great use in industries for its unique lower pungency [66]. In the molecular structure of capsinoids there is an existence of ester functional group in association with vanillyl group along the carbon chain [94,95]. They are present at a very low concentration (30 µg/g^−1^) in fruit [97,98]. Besides, the pharmaceutical applications of capsinoids are relatively low and are mainly focused on capsiates. In addition, capsiate are alike to capsaicin in terms of prevention of gastrointestinal and cardiovascular diseases, pain relief and serves as a potent anticarcinogenic agent. This unravels the fact that capsiate exhibiting itself is a lesser toxic substitute for capsaicin [95,99].

#### 4.4.5. Flavonoids

It is a set of chemical compounds that is based on a structure of 15-C atoms, with more than 7000 secondary plant metabolites [100,101,102]. Each of this secondary metabolite is of low molecular weight sharing a common skeleton structure with diphenylpropanes (C_6_-C_3_-C_6_), along with two phenyl rings (A and B) bridged by a C ring of heterocyclic pyran. These groups of compounds are well known for their antioxidant properties, bioactive properties and as protective agent against hypertension, inflammation, arthritis and AIDS [101,102,103,104]. Owing to these characteristics flavonoids are now much more preferred in several branches of industries especially pharmaceutical, clothing and food [104,105,106]. They are profoundly known well for their chromogenic properties as well. Flavonoids are responsible partially for imparting colors especially carotenoids in plants [100]. Flavonoids can remove reactive oxygen species. Such property depends on the redox potential of their hydroxyphenolic groups along with their structural similarity among different structures. Flavonoids are classified with respect to the degree of unsaturation (with regard to central heterocyclic ring) and the oxidation state. The diversity in the structures of flavonoids exists due to the replacement of aromatic rings A and B by methoxy, hydroxyl groups in amalgamation of glucosides by extensive conjugations. Some of the primary structures of flavonoids include flavanones, flavanols, iso-flavones, flavones, chalcones and antho-cyanidins [71]. In chilli fruits, some quantified primary flavonoids include catechin, epicatechins, luteolin, rutin, kaempferol, quercetin and myricetin [107,108,109,110,111,112,113,114,115,116]. Flavonoids are mostly accumulated in the peel. They occur in chilli in conjugation with *C*-glycosides and *O*-glycosides derivatives [117]. Within *Capsicum* genus, quercetin and luteolin are ascribed to be the major flavonoids, which account approximately 41% of the total flavonoids in chilli [63,72,118,119,120]. They are found majorly in their hydrolyzed form. The flavonoids containing in chilli pepper is mainly attributed to the amalgamation of above cited compounds, which occurs due to acid hydrolysis with other conjugate flavonoids [97,118].

##### Factors Influencing Flavonoid Concentration

Depending on the landraces, genotypes, species and commercial varieties, the actual concentration, metabolism, biochemical synthesis of flavonoids varies in chilli fruit upon interacting with agro-ecological conditions and environment [119,121]. Studies revealed that the concentration of flavonoids depends upon the maturity of fruits [122,123,124]. Reports revealed that total flavonoid content was found reduced from the immature to mature phase of fruit [122]. Furthermore, higher flavonoid concentration was observed from initial to intermediate (breaker stage) phases than either of red ripe phase and green mature phase [125]. The fact was found consistent with other studies pertaining to flavonoid content. Reports revealed that immature green fruits possess 4 to 5 times flavonoids to that of mature fruits [72,120,122,126]. Flavonoid concentration decreases upon ripening in *C. Chinense* Jacq. (Habanero) [127]. However, the behavioral pattern is found different in peppers that ripen in different colors i.e., flavonoid and quercetin content was found in green fruits as compared to red fruits [122]. The concentration and composition of flavonoids defines the organoleptic features of chilli fruit. In addition, such differences in composition and concentration of flavonoid content are directly related to the consumer preferences and usage [122]. In addition, variations in flavonoid concentration are attributed to analytical parameters such as preparation of samples, quantification and extraction methods. A study by [117] demonstrated that 3 glycosylated flavonoids (luteolin 6-C-glucoside, apigenin 6-C-glucoside-8-C-arabinoside and luteolin 6,8-di-C-glucoside) possessing an ability to scavenge superoxide radicals that are generated through different enzymatic and non-enzymatic processes rendering antioxidant effects. Such effective antioxidant properties can effect nutraceutical and functional properties to human health.

#### 4.4.6. Capsaicinoids and Flavonoids on Human Health

Chilli fruits, in addition to sensory characteristics on foods, also provide nutritional benefits owing to its chemical composition of vitamins, minerals, carotenoids, capsaicinoids and flavonoids [128,129,130,131]. Till date there is no concrete evidence with regard to the beneficial and antagonistic health effects on consuming chilli. In a recent study by Chopan and Littenberg [128] revealed that there is no undeviating way of correlating mortality rate and high consumption of chillies. In another study by Lv et al. [129] unraveled the fact that there is direct association between mortality due to respiratory or cardiac diseases, diabetes, cancer and consumption of fresh or dried chillies. Moreover, the genus *Capsicum* is acclaimed for antimicrobial, anticoagulant, antidiabetic, anticholesterolemic and antioxidant properties [132,133].

Capsaicinoids including hydrocapsaicin and capsaicin aids in carcinogenic activities and helps in eliminating ROS and thereby imparts antimutagenic, anticarcinogenic and prevention properties [134]. Cancer is globally one of the prime causes behind mortality rate. The etiology of this disease is implied on the oxidative stress, which stems out from the disparity between production of ROS and defense system of the cells [135]. Such Reactive Oxygen Species (ROS) has the capacity to de-regulate (redox) homeostasis and cancerous tumors that by inducing signaling networks, which results in tumors [136]. ROS regulate wide range of networks in cell signaling, which is achieved through kinases, enzymes, proteins, transcription factors viz., STAT3 and NF-κB and other hypoxia-inducible factor all of which induces in the development of growth of cancerous cells [137]. Capsaicinoids thus have the properties to reduce the multiplication of cancerous cells and thereafter the migration and initiation of apoptosis [66]. Further in vitro and in vivo study trials had revealed that capsaicinoids can reduce the growth and multiplication of prostate cancer. In addition, capsaicinoids stimulates JNK (c-Jun N-terminal kinase), signaling of MAP protein kinase and ERK (extracellular signal-regulated protein kinase) which exhibits anti-proliferative effects [137]. Furthermore, capsaicinoids initiates apoptosis by alleviating p53, p21 and Bax mediated through androgen receptors named AR-positive (LNCaP) and AR-negative (PC-3, Du-145) [138]. Capsaicinoids can stimulate the transformation of cholesterol into bile acids through increased secretion of bile acids in feces and expression of the gene named cholesterol 7-hydrolase [139].

Capsaicin (major component of capsaicinoids) serves as anti-tumor agent in treating gastric and pancreatic cancer and induces apoptosis in gastric cancer [138,139,140] by generating ROS via mitochondria, causing draining out of intracellular antioxidants [140]. Reports revealed that capsaicin can promote weight loss upon chilli consumption by stimulating lipolysis [140,141]. This results in alleviating hydrolysis of triacylglycerol especially in adipocytes and reduction in intercellular lipids [141]. Such effects are mediated through gene regulated association of some catabolic pathways of lipid, especially HSL and CPT-Ia and UCP2 gene, which is involved in thermogenesis [142]. Further studies revealed that capsaicin has the potentiality to remove visceral fat and prevents obesity.

Dihydrocapsaicin (another component of capsaicinoids) exhibited profound antibacterial potentiality against the bacteria *Helicobacter pylori*, which is associated with gastric cancer [136]. Further studies on this aspect demonstrated that oral consumption of quercetin in vivo reduces sepsis from *Helicobacter pylori* and simultaneously reduce lipid peroxidation and inflammatory response [136,139].

Flavonoids, important component of the genus *Capsicum*, provide not only modulatory action on the signaling of lipid kinases and proteins in cell but also aid as conventional antioxidant hydrogen donors [140,143,144]. The flavonoid, apigenin, has profound effect against breast cancer cells and served as an inhibitor of protein kinases [145]. In a study it was reported that the alygone of quercetin renders defensive effect on nerve cells against nephrotoxicity instigated oxidative stress involving Alzheimer’s disease [132].

### 4.5. Overview on Antioxidant Activities

Antioxidant, a term in general, defined as aniota that has the capacity to delay or prevent the significant way of oxidation of easily oxidizable materials, such as fats or lipids [146,147,148,149]. Not only the lipids that undergo oxidation but it has also been well documented that DNA and protein are very much prone to oxidation [150,151]. In short, antioxidants are the substances that are proficient enough to repair systems. Such system comprises of Fe-transporting proteins that reacts especially with oxidizing agents by inhibiting a specific oxidizing enzyme. Having said that, there is no apt definition worldwide to the term “antioxidant”. Scientists have great deal of attention in plant-derived antioxidants. Capsaicin has the ability to prevent or delay the substrate from getting oxidized [152]. Such antioxidant potentiality of capsaicin was justified later wards by Oyagbemi et al. [148]. The study revealed that capsaicin can prevent carcinogenesis by blocking the translocation of activator protein-1 (AP-1), activator and signal transducer of the transcription (STAT 3), nuclear factor kappa β (NF-kβ). Besides, through generation of ROS capsaicin can arrest further cell cycle and induce apoptosis [148]. Studies have further revealed that capsaicin is potential enough to impede radiation-induced biochemical transformations. Such alterations include lipid peroxidation and oxidation of protein [153], which suggests that physiologically capsaicin can serve as ‘radio-protector’. As mentioned earlier, capsaicin has ability to counteract effects of Reactive Oxygen Species that are associated with cellular metabolic processes causing neurological disorders, cancer, diabetes and cardiovascular disorders [154]. By consuming chilli pepper, detoxification of ROS is possible by several enzymes present in dietary components.

The antioxidant properties in chilli can be attributed and judged by presence of several assaying systems such as dimethyl-4-phenylenediamine assay (DMPD), 2, 2-diphenyl-1-picrylhydrazyl assay (DPPH), 2, 2′-azino-bis[3-ethylbenzothiazoline-6-sulphonic acid] assay (ABTS), Ferric ion reducing antioxidant power assay (FRAP), Cupric reducing antioxidant capacity assay (CuPRAC) [155]. Furthermore, vitamin C, E, anthocyanin and phenolic compounds in chilli pepper possess antioxidant property [120,156]. Phenolic compounds in chilli donate H atoms that neutralize and scavenge radicals inside the body [156]. Medina-Juarez et al. [118] reported that there exists an interdependence between phenolic content and antioxidant potential in chilli pepper. Castro-Concha et al. [155] reported higher levels of carotenoids, ascorbate and glutathione attributed to higher antioxidant activity in *Capsicum chinense*. Such antioxidants, glutathione and ascorbate contribute to Halliwell Asada Cycle, which detoxifies H_2_O_2_ produced in chloroplast without aid of catalase enzyme [157]. According to the findings of [138] capsaicin has been reported to provide synergistic effect with certain therapeutic drugs especially 5-fluorouracil (5-FU). Such drug enhances sensitivity to cholamgiocarcinoma (CCA), a multi-drug resistance type that suppresses cancer cell growth [158,159,160]. The antioxidant activities of capsaicin in chilli are comparable to the synthetic antioxidants especially butylatedhydroxyl toluene (BHT) and butylated hydroxyl anisole (BHA) [159]. Furthermore, intervention studies of capsaicin for both pure capsaicin and microemulsions exhibited higher inhibitory potentialities than synthetic BHT [161]. Such formulations can preferably be used as natural preservatives in meat.

### 4.6. Anti-Cancerous Perspectives

Phytochemicals especially capsaicin present in chilli can reduce carcinogenesis and trigger apoptosis in cancerous cells especially in the tissues of skin, colon, bladder, breast, prostate and lung [162,163,164]. Reports revealed that capsaicin has far-reaching anti proliferative activity on human prostate cell in culture. It may act mainly in two different ways especially in case of prostate cancer, direct and indirect pathways. Direct pathway is aided through (i) engendered destruction of primary prostate cancer cells (ii) holding back the expression of prostate–specific antigen (PSA) and (iii) Repression of PSA transcription, which aids in plunge of PSA levels [134]. Another direct pathway of capsaicin that retards the cancer cell growth is antagonistic effect of coenzyme Q controlling the electron transport. Indirect pathway is mediated through interaction of capsaicin with TRPV1 receptor cells resulting in accumulation of Ca^2+^ cations into the cancerous cells leading to production of apoptosis [164]. Thus, role of Ca^2+^ at intracellular level have considerable effect on cancer cells including TRP channels leading to intensification of amount of ROS, apoptosis in cancer cells and depolarization of mitochondrial membrane [164,165]. Studies reported that capsaicin can choke migration of breast cancer cell and its metabolites. Metabolites such as phenoxy radicals, which are highly reactive in nature, can even attack DNA molecules and stir up mutagenicity and fatal transformation [166]. Other line of evidence revealed that capsaicin can inhibit nuclear factor-kappa aka NF-κ activation along with tumor necrosis factor-alpha aka TNF-α in prostate cancer cells [134]. Similar anticancerous effects were observed under different conditions pertaining to hepato-cellular carcinoma [132], colon cancer [162], leukemia [164], gastric cancer [165,166] and lung cancer [167] (Table 2). Fruit extracts of *Capsicum chinense* when applied to cell lines of HepG2 exhibited inhibition of proliferation of hepatocellular carcinoma [63]. Capsaicin can actually bind with ATP generating coenzyme Q and thereby inhibits electron flow in mitochondria through generation of ROS and triggering apoptosis [168]. This ROS thus generated helps in inducing apoptosis pathways and disrupt mitochondrial membrane in pancreatic cancer cells [169]. Research revealed that capsaicin can drastically reduce the seizure alongside migration of cholangiocarcinoma HuCCT1 cells by disabling NF-κB/p65 signaling pathway including expression of MMP-9 [168,169]. Inhibition of nuclear factor kappa-[light-chain-enhancer of activated] B cells (NF-κB) network by capsaicin can simultaneously minimize carcinogenic affects and escalate apoptosis [170,171,172,173]. In a study, Zhang et al. [174] demonstrated that capsaicin through stimulation of TRPV1 can block nuclear relocation of proliferating cell (nuclear) antigen, thus it proves that relation between capsaicin and TRPV1 could be a viable option in bladder cancer treatment. Lin et al. [175] reported that capsaicin can be used in triggering the apoptosis pathway in human κB cancer cells via agitation of mitochondrial membrane including caspase signaling network. Studies revealed that on consuming capsaicin orally can lower down pancreatic cancer, while working in mice [171] and upon consuming capsaicin in diet for 16 months reports revealed that tumor rates were lower particularly in liver cancer cells [132]. Studies further revealed that capsaicin can repress chemically induced skin cancers in mouse models [172,176,177]. Studies related to skin cancer revealed that capsaicin can down-regulate *Bcl-2* (B-cell lymphoma 2) expression and bringing about autophagy in B16-F10 lines (acquired from *Mus musculus* skin melanoma) melanoma cells [178,179,180]. Human cells upon culture with exogenous capsaicin exhibited autophagy [181] apoptosis and inhibition of cell metabolism [182,183,184]. Further experiments conducted by [185,186] by using acquired human cutaneous squamous cell carcinoma (SCC) lines revealed that capsaicin has therapeutic and prophylactic potential in inducing apoptosis and modulating the epidermal growth factor receptor (EGFR) in skin cancers [187].

In addition, capsaicin has proved to induce apoptosis through oxidative stress in leukemia cells [174]. Furthermore, in vitro studies revealed that capsaicin can inhibit the growth of blood cancer by inhibiting human T-cell leukemia virus type 1 transcriptional transactivator aka Tax-proteins followed by increasing NF-κB inhibitor with α triggering apoptosis by blocking cell cycle. This substantiates the role of capsaicin as a chemopreventive drug against leukemia [173,191]. A study by Amantini et al. [184] reported that capsaicin has potential effect in treating glioma cancers cells of brain by binding potential TRVP1 in glioma cells triggering autophagy by down-regulating *Bcl-2* expression. Research round the world also depicted that capsaicin can impose inhibitory effects on cancerous cells in digestive systems, tongue cancers by inducing the expression of caspase-3 and caspase-9 activities [186,192]. Capsaicin can also mediate ROS activity and induce apoptosis in pancreatic cancer cells suggesting potential use of chilli in treating pancreatic cancer [137]. Lee et al. [191] reported that cell proliferation in colorectal cancer cell lines can be suppresses by capsaicin by restricting the expression of transcription factor 4 (TCF 4) and inhibiting interplay of β-catenin and TCF-4. In addition, capsaicin has the potentiality to lessen the risk of lung cancer in both cancer cell experiments and mouse models by holding back response of E2F (a transcription factor gene in higher eukaryotes) and genetic expression, which discourages its proliferation [188]. In mouse models, capsaicin minimizes the risk of cancer by stabilizing mitochondrial related enzymes using benzopyrene [191,192]. Furthermore; the anticancerous property of capsaicin is also seen in nasopharyngeal related carcinoma by inducing apoptosis mechanisms in NPC lines as reported by Ip et al. [189].

### 4.7. Anti-Obesity Activities

Worldwide, obesity is recognized as complex medical condition due to its predominant effects on finance, morbidity and mortality. The disease is the result of interaction effects of sedentary lifestyle, environmental, genetic and dietary effects leading to increased body fat mass. In recent past years, prevalence of obesity has been reported to be the highest in the age category ranging from 50–59 years, revealing an overweight of 60.2% of the population in Malaysia [10]. Alongside the adult population, over the years such increasing trend is observed in teenage group as well. In latest study by NHMS, one in every two adults in Malaysia was obese and females ranked in the highest category [10]. Recently a study by [40], *C. annuum* contributed to thermogenesis, on application of capsaicin on 3T3-L1 adipocytes lipid content at intracellular level was found to be decreased and thus involved in thermogenesis. In a study by Kang et al. [159], it was affirmed that capsaicin in dietary form could lessen the number of liposaccharides (LPS) content and high-fat diet provoked CLGI related anti-obesity factors. They further reported that capsaicin can also lower down metabolic malfunction in obese or diabetic KKAγ by increasing dissemination of adiponectin and its receptor in mice [159]. Lee et al. [191] reported that lipid accumulation was found decreased in epididymal and mesenteric adipose tissue on application of 0.075% capsaicin in high fat induced obese mice. Furthermore, serum level of triglycerides, cholesterol and glucose was decreased on application of capsaicin in mice. Studies also revealed that capsaicin has anti-proliferative activity in preventing the 3T3-L1 pre-adipocyte and down-regulates the transcription factors, especially PPARγ and thus helps in maintaining body weight [138,144]. Van Avesaat et al. [192] reported that capsaicin has the ability to induce satiety in a short time span, which in turn decrease calorie intake leading to weight loss. In a latest study by Baskaran et al. [193] revealed that upon feeding capsaicin (1µM) heightened the thermogenic and adipogenic proteins in brown adipose tissue (BAT) by stimulating SIRT-1/PRDM-16 dependent methods exhilarating anti-obesity effects.

### 4.8. Cardiovascular Roles

The cardio-vascular system in our body is comprised of wide range of sensory nerves sensitive to capsaicin, which plays an effective role in managing cardiovascular function by releasing several neurotransmitters especially Calcitonin gene-related peptide aka CGRP and others including substance P. CAPS has potentiality to provide beneficial effects in cardiovascular system. Peng and Li [194] reported that capsaicin can stimulate release of CGRP by activating TRPV1 benefitting cardiovascular function. Studies by Adams et al. [195] reported that dihydrocapsaicin and capsaicin has potentiality to block platelet aggregation and clotting factors VIII and IX. This helps in lowering the incidence of cardiac diseases. Furthermore, capsaicin can insert itself into the plasmalemma by altering fluidity of the membrane and maintaining its ionic permeability [196]. In addition, Harper et al. [197] reported that capsaicin was capable enough to release calcium cations from intracellular platelets containing TRPV1, which contributed successively to ADP triggering platelet aggregation. Further studies revealed that capsaicinoids especially dihydrocapsaicin can very low-density lipoprotein cholesterol aka LDL-C, low-density lipoprotein cholesterol aka LDL-C, plasma cholesterol as well as C-reactive protein aka CRP, inflammatory cytokines such as interleukin 1 beta aka IL-1β, IL-6, tumor necrosis factor-alpha aka TNF-αand triglycerides. Findings further confirmed that through plasma sterol analysis capsaicinoids can lower plasma cholesterol levels thereby reducing cholesterol absorption. Ahuja et al. [198] reported that on consuming chilli fruits by adult men and women regularly for 4 weeks (3 µg per day approximately) can provoke resistance of serum lipoproteins to oxidation. Manjunatha and Srinivasan [199] reported that capsaicin with concentration of 15 µg/kg of body mass of high-fat-fed rats can reduce lipid peroxide and serum total cholesterol level. The antioxidant property of capsaicinoids attributed to elevated HDL levels and can assiduously enhance the CRT i.e., Reverse Cholesterol Transport pathway, resulting in arresting atherosclerosis and simultaneously promoting cholesterol inflow in THP-1 macrophage acquired foam cells [200]. These above-mentioned statements thus prove that anti-oxidant property of capsaicinoids in cardiovascular, coronary heart diseases and others.

### 4.9. Anti-Hyperglycemic/Antidiabetic Activities

Capsaicin has a potent role in reducing glucose and insulin levels in humans. By 2025, studies suggested that number of diabetic patients may rise up to 300 million [201] and thus economically efficient treatments are necessary. Chilli pepper behaves as storehouse of new drugs, which includes α-amylase inhibitor and α-glucosidase inhibitor, which are potent enough to impart antidiabetic activity [201]. On consuming chilli, D-glucose absorption in the intestine is lowered, which could further be utilized in post-prandial rise of glucose and carbohydrate degradation [95]. Studies demonstrated that capsaicin has an effect in carbohydrate metabolism [202,203,204,205,206,207,208,209]. Research on mouse models by Kang et al. [210] revealed that capsaicin on altering gene expression can lessen hyperglycemia. The rate of carbohydrate metabolism were also utilized across human and rats to obtain deep insight in judging anti-diabetic effects of chilli. On administration of capsaicin to rat models through the process of thermogenesis reactions lowers blood glucose levels [88]. Besides, *C. frutescens* also plays pivotal role in increasing blood insulin level in rat models because of affinity of insulin towards glucose [203]. This depicts involvement of *C. frutescens* in treatment of diabetes. Recent advances in research pertaining to diabetes by [210], it was revealed that receptors involving TRPV1 play a pivotal partin progression of diabetes type 1 and 2. This progression and development of the diabetes is modulated by capsaicin expressing sensory neurons by ablating TRPV1. Such permanent exclusion of TRPV1 induced by capsaicin decreases insulin resistance, islet infiltration and β-cell stress [207,208,209]. This proves that depletion of TRPV1 expressing neurons prevents diabetes, which is genetically predisposed to type 1 diabetes in mice. Fattori et al. [205] further used Zucker diabetic fat (ZDF) rats for studying various aspects of type 2 diabetes in human and revealed that exclusion of TRPV1 by capsaicin expressing sensory fibers in islets of Langerhans decreased plasma glucose levels, increased insulin secretion and enhanced glucose tolerance. Fattori et al. [205] further demonstrated that on consumption of 10 µg of capsaicin per kg of body mass of mouse models could weaken the activation and augmentation of auto-reactive T cells especially in pancreatic lymph nodes (PLNs). This helps in protecting mice from diabetes. The above studies prove that capsaicin heightens TRPV1 activity, which is important in regulating the activity of islets. A study by Weitz et al. [206] revealed that microphages in islet are capable enough to impart immunological destruction of type 1 diabetes and simultaneously helps in bringing about inflammation in type 2 diabetes.

### 4.10. Anti-Inflammatory and Pain Relieving Activities

Out of several phytochemicals in chilli, capsaicin has the ability to confer anti-allergic and -inflammatory activities [211]. The pigment anthocyanin also possesses anti-inflammatory activity [208]. Capsaicin can reduce the inflammatory responses provoked by antigens [209] and simultaneously inhibits dissemination of pro-inflammatory cytokines [210]. Such cytokines includes interleukin 1β aka IL-1β and tumor necrosis factor-α aka TNF-α. Capsaicin can induce inflammatory effects in adipose tissues indicating potential applications as an analgesic [212]. Studies revealed that nordihydrocapsiate (a type of capsinoid) and capsaicin can prevent T cell activation at earlier stage, such events includes NF-κB activation [213]. Furthermore, capsaicin can reduce inflammatory heat, side by side, reports revealed its ability to provide pain relief pertaining to noxious hyperalgesia, which controls the secretion of neurotransmitter causing pain [214]. Studies further demonstrate that capsaicin is capable enough in reducing neuropathic pain [213], as a soothing agent for oral mucositis [215] and bladder pain [216].

### 4.11. Anti-Microbial Activities

Spice chilli has been widely used in preserving food since time immemorial [217]. Compounds especially capsaicin that imparts antimicrobial activity along with pungency were observed to be present in *Capsicum* sp. In addition to pungency, anthoscyanin pigment in chilli also possesses antimicrobial activity [218,219]. Several studies revealed antimicrobial activity of chilli against pathogens especially *Clostridium sporogenes*, *Streptococcus pyogenes* and *B. subtilis*, *Candida albicans*, *Escherichia coli*, *Sarcinalutea*, *Pseudomonas aeruginosa* [217,218,219]. Jones et al. [220] observed bacterial activity of capsaicin against *Helicobacter pyroli* causing gastric disorders where inhibitory concentration of capsaicin at bare minimum was found to be 10 µg/mL, which was later confirmed by Zeyrek and Oguz (2005) [221]. Zeyrek and Oguz [221] further observed that capsaicin also exhibited inhibitory activity against wide spectrum of microbial strains. Nascimento et al. [222] observed that the extracts of phytochemicals from *C. frutescens* especially dihydrocapsaicin, capsaicin and chrysoeriolhad antimicrobial activity against *S. aureus*, *E. coli*, *Enterococcus faecalis*, *B.subtillis* and *Klebsiella pneumonia*. In another study by Kurita et al. [223], the antimicrobial capacity of capsaicin was judged with the aid of DNA microarray technology where toxic effect against yeast cells were observed and exerts pleiotropic network of drug defiance, which exhibits genes in relation to membrane biosynthesis and osmotic stress.

### 4.12. Anti-Clotting Activity

Phytochemicals in chilli have proven healing effects in wide range of diseases. Studies in mouse models unraveled that capsaicin can inhibit platelet coagulation by withstanding clotting factors VIII: C and IX [195,197]. Studies by Wang et al. [224] demonstrated that capsaicin is more effective than aspirin in altering acute pulmonary thromboembolism. Furthermore, capsaicin can prevent platelet formation by stabilizing membranes of RBCs by impeding the enzyme phospholipaseA2 aka PLA2 as reported by Wang et al. [224]. Therefore, capsaicin has the potential to serve as anti-clotting agent [225].

### 4.13. Anesthetic Activities

The capsaicinoids (especially capsaicin) have the ability to provide relief to a wide range of pain disorders in human and are used to prevent several clinical situations related to pain, which includes contact allergy, shingles (Herpes zoster), post-mastectomy syndrome, postsurgical neuromas, cluster headaches, urological disorders, diabetic neuropathy, pruritis, and many others [153]. Friedman et al. [226] reported that proliferation of tumor occurs well under in an acidotic environment, which elevates upon interference with TRPV1 antagonists relieving pain symptoms.

Borbiro et al. [227] revealed that on TRPV1 activation by capsaicin piezo proteins can be obstructed. Such obstruction arises due to Ca-dependent incentive of phospholipase Cδ aka PLCδ, which lowers down phosphor inositides. Administration of such phosphoinositides in the cytosol caused by purged patch clamp, which reduces inner current of Piezo channels and thus reverts inactivation [228]. Furthermore, capsaicin has the capacity to trigger non-neural TRPV1, which stimulates the liberation of prostaglandin E2 (PGE2), IL-6 and interleukin (IL-) 8 [229]. In addition, direct pharmacological desensitization of plasma membrane and inactivation of voltage-gated Na^+^ channels by TRPV-1 receptors can result in impromptu reduction on neuronal responsiveness and excitability. Anand and Bley [230] further added that excessive concentration of capsaicin (than required) to TRPV1 can result in antagonist effect in mitochondrial dysfunction by inhibiting openly electron chain transport. Further findings had reported that capsaicin can interact with nerve endings, which is transgerminal in nature, and release a neurotransmitter called substance P. Substance P is a neuropeptide, which follows amino acid sequence as Arg-Pro-Lys-Pro-Gln-Gln-Phe-Phe-Gly-Leu-Met [RPKPQQFFGLM]). Studies by Yang and Du [231] revealed that capsaicin can stimulate in release of substance P especially in arthritis and after the recurrent application capsaicin can drain out neuron of substance P and prevents its re-accumulation. Furthermore, it has been observed in rheumatoid arthritis (RA) that sensory afferents, which are highly capsaicin sensitive and densely innervate the synovium and the articular capsule. Thus, capsaicin plays a potential role in anesthesia in clinical conditions.

### 4.14. Asthma and Rhinitis Treatment

Systematic studies by Van Gerven et al. [232] revealed that capsaicin can employ its therapeutic action TRPV1-substance P, which produces nociceptive signaling pathway in the nasal mucosa. In addition, a recent study had exhibited that capsaicin helps in desensitization of sensory nerves relieving NAR and IR symptoms for approximately 9 months [211]. Further study revealed that such NAR or IR was correlated with an amplified dissemination of substance P levels in nasal secretions and TRPV1 in the nasal mucosa. A study by Van Rijswijk et al. [233] reported intranasal applications of capsaicin in imparting potential medical benefits in rhinitis type of disorders. Capsaicinon application, provokes initial irritation to the area applied but with passage of time it progressively desensitizes the sensory neural fibers lowering nasal hyper-responsiveness.

### 4.15. COVID-19 Treatment

Presently, the importance of improving the immune system has turned a vital issue since the outbreak of SARS-CoV-2 and capsaicin has potent role in defending immune related syndromes. TRVP1 is a profound modulator of neuroinflammation and in treating autoimmune diseases. TRVP1 receptor serves as a target of immunomodulatory networks in immune cells particularly on macrophages [227,232] and is also known to serve indirectly in unmyelinated nerve fibers. On oral administration of capsaicin of 10 µg concentration instead of producing inflammatory cytokines it produces IL-10 in a mouse model having autoimmune diabetes [233,234,235]. In addition, a person when affected by COVID-19 was reported to possess lower lymphocyte level and NK (Natural Killer) cells. A recent study reported by Prompetchara et al. [236] on SARS-CoV-2 revealed that the virus can enter the cell through an enzyme called angiotensin-converting enzyme-2 aka ACE-2, which is mostly identified by a receptor called Toll-like receptor-7 aka TLR-7. Activation of TLR-7 leads to the secretion of IL-12 and IL-6 and the production of α-interferon. In addition, activation of such TLR-7 results in production of CD8+ specific cytotoxic T cells via CD4+ T cell form and production of antibody. Further study by Giamarellos-Bourboulis et al. [237] described that Severe Respiratory Failure (SRF) was shown by more than one-fourth of this virus infected patients and can later develop Macrophage Activation Syndrome (MAS). This could finally lead to immune dysregulation in subjugation with reduced expression of HLA-DRon CD14 (a type of monocyte). Such process happens by triggering of IL-6, which liberates disproportionately and monocyte hyperactivation including an intense lymphopenia. All this combined cases are aided by proliferation of viral produce and a storm of cytokine called Cytokine Release Syndrome (CRS) are observed. In majority of cases causing death rates generally observed as 28% of fatal COVID-19 cases. Moreover, in most of the clinical cases, multi-organ damage including failure of hepatic, cardiac and renal systems causing death eventually [238,239].

Studies by Janda and Iadarola [240] suggested that TRVP1 has potent role in the matter of prognosis especially in case of viral infections, such as SARS-CoV-2. As per Nahama et al. [241], TRPV1, which expresses innervations in combination with the virally aided hyper-inflammation in COVID-19 cases in mature patients, might be the essential cause of the lethality of the disease. Further findings suggested that there exists a strong correlation between a strong immune response and SARS-CoV-2 and chain of reaction pertaining to immune system. On interference of TRVP1 signaling virus infected patients might lower down ARDS, which is popularly called Acute Respiratory Distress Syndrome. Thus, by silencing TRPV1 positive nerve fibers can limit the disease progression from mild to acute respiratory distress. Studies by Chen et al. [242] revealed that viral protease 3CL-protease, a protein is solely responsible for virus replication and blocking such enzyme can inhibit the virus from replicating in human body. Such unique enzyme possesses three functional domains. The enzyme with such a structure provides ample scope to different compounds to bind and inhibit it, inhibiting virus from replicating further Kumar et al. [243]. Major international research by Kadil et al. [244] suggested some possible treatments against COVID-19. They elucidated possible pathways of 3CL-Protease enzyme activity inhibition alongside scouting the use of wide range of chemical compounds especially remdesivir, azithromycin, indinavir, ritonavir, oseltamivir and others particularly used in treating viral diseases.

Capsaicin along with other 33 compounds has the capacity to dock at molecular level towards 3CL-protease [245]. Further, capsaicin can form hydrogen bonds especially at 4 non-identical sites of 3CL-protease enzyme (PRO168, HIS163, CYS145, THR190) binding with free energy of −8.15 kcal/mol (estimated). Another study by Barros et al. [246] revealed that being on the active site capsaicin has the ability to interact with GLU166 hydrogen bonds but lacks the ability to exhibit good binding ability towards the said enzyme. Further studies revealed that molecular docking of RdRp, which is popularly called RNA-dependent RNA polymerase, a key enzyme replication of virus with capsaicin binding at free energy of −7.3 kcal/mol, which is almost similar to remdessivir drug having free energy binding of −9.0 Kcal/mol [247] and a target to search out actual therapeutic agents for COVID-19. Thus, these above studies ascertain that capsaicin alone is a potent enough antiprotease drug to combat SARS-CoV-2 virus replication.

## 5. Conclusions

Chilli fruit and its associated parts possess epoch-making applications and a wide spectrum of bioactive compounds in different domains involving agriculture, food, pharmaceuticals, medicines, and cosmetics. The by-products of this plant have also proven beneficial in the textile industry. At present, most of the studies related to chilli are mainly focused on characterization and synthesis of bio-active constituents, including possible extraction methods using different solvents. The trend is gradually shifting from use of synthetic ingredients to natural ones. Such paradigm shift has resulted in utilization of chilli for a wide range of applications, signifying the potentiality of the crop.

From the above discussion, it is clear that capsaicin and other phytochemicals present in chilli play a vital role in different medicinal benefits, including anticancer, antidiabetic, anti-obesity, antibacterial, antifungal, anti-clotting, and anti-inflammatory activities. The phytochemicals involved along with their corresponding molecular mechanisms have been characterized to exhibit their modes of action. Such revelations were confirmed using animal models and human cancer cells. The handful of information that is currently available with meticulous research on biomedical perspectives of chilli pepper by skillful harmonization with *omics* technology, IT, computerized drug designing, molecular docking and other sophisticated tools can be used to develop drugs to combat cancers along with other important ailments. Thus, chillies can serve as reservoir of potent drug formulations, which could precisely upgrade and enrich our pharmaceutical industries for one or more medical and pathological conditions and towards human wellbeing.

## 6. Future Directions

Research works are essentially required to characterize and explore essential bio-active compounds from chilli peppers in diversified geographical conditions and to develop novel strategies and techniques for enhancing extraction efficiency, enhancing bioactive compound content and broadening the horizon of applications of these functional components. In addition, phytochemicals acquired from chilli for food preservation are very promising and would open many closed doors for further research of other non-pungent derivatives that deserve special attention. Studies to date reveal that the effect of capsaicin looks promising in the animal and laboratory studies but remains unavailable in terms of human trials; thus, wide scale human studies are needed to assure health benefits. Quite a number of appalling adverse effects including ulcers, gastric pain, eye tearing, and sweating are related to the pungent nature of capsaicin upon oral consumption, which may overshadow its use as a therapeutic agent and might make any clinical trial useless. Approaches such as iontophoresis, hydrogel formations, encapsulation in liposomes, formulations based on nanoparticles should be inculcated in order to make capsaicin bioavailable. Furthermore, use of second-generation capsaicin mimetics would be highly convincing in simultaneously provoking higher pharmaceutical activities and suppressing adverse effects of capsaicin, thus, capsaicin mimetics and analogs of capsaicin which might form the basis of medical treatments against wide spectrum of human disorders and diseases.

## Figures and Tables

**Figure 1 molecules-27-06380-f001:**
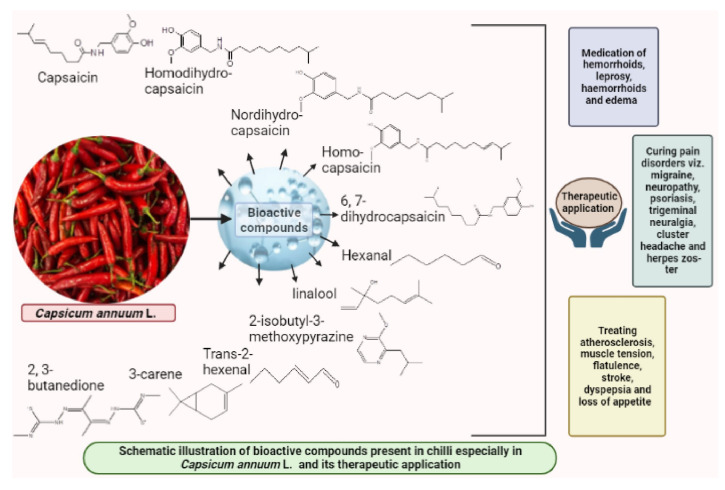
Bioactive compounds present in chilli, their structure and therapeutic applications.

**Figure 2 molecules-27-06380-f002:**
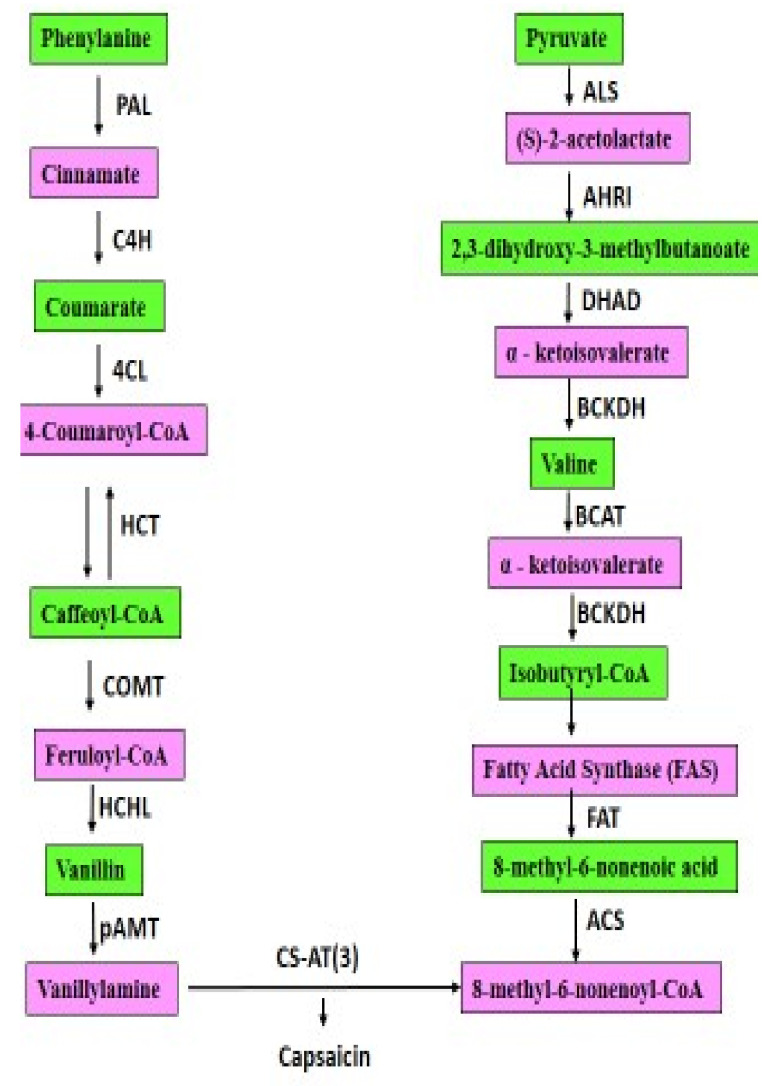
A model pathway depicting biosynthesis of Capsaicin.

**Figure 3 molecules-27-06380-f003:**
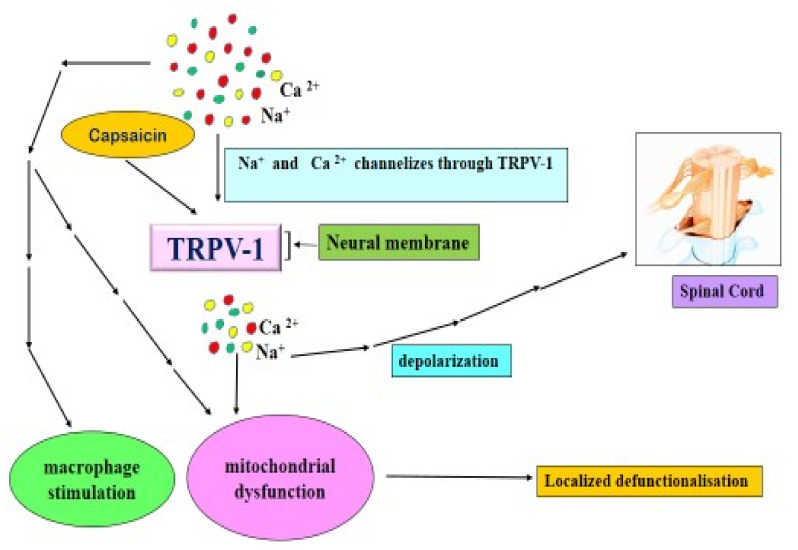
Effect of capsaicin on TRPV1 receptor [53,54,55].

**Figure 4 molecules-27-06380-f004:**
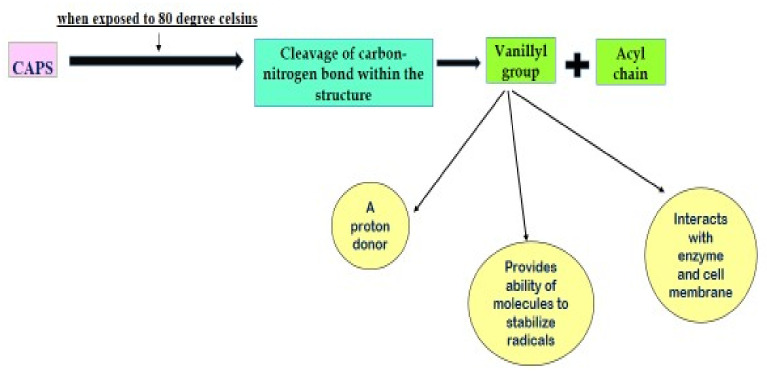
Bioactivity of CAPS attributed to Vanillyl group present in the structure [65,66].

**Figure 5 molecules-27-06380-f005:**
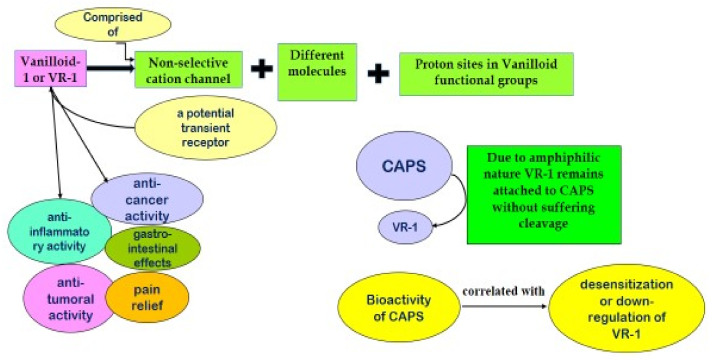
Interaction of CAPS and VR-1 [65,66].

**Table 1 molecules-27-06380-t001:** Nutritional contents of chilli fruits across species (per 100 g of edible portion).

Nutrients	Peppers, Hot Chilli, Green (Raw) {a}	Peppers, Sweet, Green (Raw) {b}	Spices, Pepper, Red or Cayenne {c}	References
Carbohydrate (g)	9.46	4.64	56.63	[19,20,21]
Protein (g)	2.00	0.86	12.01	[19,20,21]
Fat (g)	0.20	0.17	17.27	[19,20,21]
Energy (kcal)	40	20	318	[19,20]
Iron (mg)	1.20	0.34	7.80	[19,20,21]
Calcium (mg)	18	10	148	[19,20,21]
Sodium (mg)	7	3	30	[19,20,21]
Potassium (mg)	340	175	2014	[19,20,21]
Phosphorus (mg)	46	20	293	[19,20]
Copper (mg)	0.30	0.066	0.373	[19,20]
Selenium (μg)	0.5	0.0	8.8	[19,20]

{a}—Botanically: *C. frutescens.* {b}—Botanically: *C. annuum.* {c}—Botanically: *C. frutescens* or *C. annuum.*

**Table 2 molecules-27-06380-t002:** Inhibition effects of capsaicinoids on diversified cells satirizing the antitumor activity.

Type of Cancer	Diversified Cell Lines	Inhibitory Effects	References
Pancreatic cancer	BxPC-3 and AsPC-1 (pancreatic cancer)	Inhibitory effects by generation of ROS resulting in induction of apoptosis	[137]
Blood leukemia	Human myelocytic leukemia (HL-60)	Inhibitory effects by induction of autophagy by caspase-3-dependent process	[164]
Human KB cancer	KB (which is derived from HeLa cell line)	Inhibitory effects by staggering cell cycle at G2/M phase causing inducing apoptosis	[175]
Tongue cancer	Squamous-Cell Carcinoma (SCC-4)	Inhibitory effects by mitochondria dependent and independent mechanisms causing induction of apoptosis	[186]
Lung cancer	NCI-H69, NCI-H82	Inhibitory effects by arresting cell cycle at GI	[188]
Nasopharyngeal cancer	Nasopharyngeal Carcinoma (NPC-TW 039) in human	Inhibitory effects by mitochondrial alteration and stress in endoplasmic reticulum causing induction of apoptosis	[189]
Hepatic cancer	HepG2 (human hepatoma)	Inhibitory effects by disruption of ROS causing induction of apoptosis	[190]

## Data Availability

Not applicable.

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
