# Peer review of "Biomedical and Antioxidant Potentialities in Chilli: Perspectives and Way Forward"

_molecules, 2022, doi:10.3390/molecules27196380_

Round 1

Reviewer 1 Report

The review is devoted to chili pepper, which contains many useful biologically active compounds that provide various therapeutic effects of this vegetable crop. The work is interesting and will be useful for a wide range of readers, but there are many comments on the design of the work.

 1. The title does not correspond to the content of the review article, since only one section has been written about antioxidant activity. It is worth changed the title of the article.

2.    Throughout the text, either there are no spaces between words and punctuation marks, or vice versa, there are spaces in words, for example: line 31 «The responsibilityof TRPV1 recepto….», line 45 «…used in cooking [1].The…», line 120 «…of chilli speciesprovides ample amount…» etc.

3.    It is necessary to decipher all abbreviations at the first mention, and not as in the case of ABTS, which is first mentioned on line 148, but the transcription of the abbreviation on line 491.

4.    There is no name and number of Table No. 1.

5.    There are no references in the text to figures and tables.

6.    The name of Figure 2 should be clearer and not just one word «receptor».

7.    Need to lead to monotony of captions to drawings, since some are bold, some are not, all names with capital letters should be.

8.    References to literature need to be redone, instead of [71,74,75,76,77,78,79] write [71,74-79] etc.

9.    Correct "in-vitro" and "in-vivo" to in vitro and in vivo

10. Present all figures in the best quality.

11. Why are the words on lines 184, 190, 579 highlighted in bold?

12. On line 154 oxygen without italics "7-di-O-α-L-rhamnopyranoside", on line 164 – in italics, correct throughout the text.

13. The names of compounds are sometimes capitalized in the text, for example on lines 226 "Furthermore, Capsaicin has...", 274 "N-vanillyl-nonanamidehomocapsaicin I, homocapsaicin II, nornorcapsaicin, N-Vanillyl-octanamide...", why?

14. After section 4.4.3.2.  is repeated again 4.4.3  and then the numbering of sections is violated.

15. Subsection 4.4.4.1 is not needed, since there is no 4.4.4.2 further, etc.

16. Line 474 "'antioxidant." correct to ""antioxidant"."

In general, the work looks slappy and the authors need to work well with the text and design of the illustrative material.

Author Response

  1. The title does not correspond to the content of the review article, since only one section has been written about antioxidant activity. It is worth changed the title of the article.

Suggested title by authors- Biomedical And Antioxidant Potentiality In Chili: Perspectives  And Way Forward

  1. Throughout the text, either there are no spaces between words and punctuation marks, or vice versa, there are spaces in words, for example: line 31 «The responsibilityof TRPV1 recepto….», line 45 «…used in cooking [1].The…», line 120 «…of chilli speciesprovides ample amount…» etc

Done in the text as advised.

  1. It is necessary to decipher all abbreviations at the first mention, and not as in the case of ABTS, which is first mentioned on line 148, but the transcription of the abbreviation on line 491.

Corrections made as advised.

  1. There is no name and number of Table No. 1.

Name given.

  1. There are no references in the text to figures and tables.

References added to figures and in tables individual references were added before

  1. The name of Figure 2 should be clearer and not just one word «receptor».

Corrected and suitable caption given

  1. Need to lead to monotony of captions to drawings, since some are bold, some are not, all names with capital letters should be.

Done as suggested

  1. References to literature need to be redone, instead of [71,74,75,76,77,78,79] write [71,74-79] etc

Done as suggested

  1. Correct "in-vitro" and "in-vivo" to in vitroand in vivo

Done as suggested

  1. Why are the words on lines 184, 190, 579 highlighted in bold?

All bold words are not-bolded wherever told to do

  1. On line 154 oxygen without italics "7-di-O-α-L-rhamnopyranoside", on line 164 – in italics, correct throughout the text.

Corrected thoroughly in the text

  1. The names of compounds are sometimes capitalized in the text, for example on lines 226 "Furthermore, Capsaicin has...", 274 "N-vanillyl-nonanamidehomocapsaicin I, homocapsaicin II, nornorcapsaicin, N-Vanillyl-octanamide...", why?

Corrections made throughout the entire manuscript

  1. After section 4.4.3.2.  is repeated again 4.4.3  and then the numbering of sections is violated.

Corrections done

  1. Subsection 4.4.4.1 is not needed, since there is no 4.4.4.2 further, etc.

Numbering removed

  1. Line 474 "'antioxidant." correct to ""antioxidant"."

Done as told

Reviewer 2 Report

rows 56-57- it is about one study in vitro, we need more to sustain the destroyng of the cancer cells by capsaicin

row 59-there are not innumerable numbers!

row 63- the results could be quite disputable, let's hope they are not quiet

row 122- the table has not number nor legend!

row 145-148- please, reconsider the phrase!

rows 201-202 - the sentence has no predicate

figure 1 has an uncomprehensible legend. what does it show? what is the connection with the text?

figure2 has an incomplete legend and has quite a poor design. please, reconsider design. also, there are no references, is it an original figure?

figures 3 and 4 are original? if not, please add references!

chapter 4.4.4.1. row 392- Factors influencing flavonoid concentration!

table 3 has no references. is it an original one?

some figures and tables are not at all mentioned in the text. why are they there?

Author Response

Point 1: Rows 56-57- it is about one study in vitro, we need more to sustain the destroyng of the cancer cells by capsaicin

Response: We have mentioned more such in vitro works in the section ‘4.6. Anticancerous perspectives’ and

there references are mentioned below-

  1. Mehmet, B.; Metin, Y.; Gulhan, A.; Omer, T.; Oruc, A. Effect of capsaicin on transcription factor in 3T3-L1 cell line. East. J. Med.2015, 20, 34-45.
  2. Mosmann, T. Rapid colorimetric assay for cellular growth and survival: Application to proliferation and cytotoxicity assays. J. Immunol. Methods.1983, 65, 55–63.
  3. Hong, Z.; Zhao, W.; Yin, Z.; Xie, C.; Xu, Y. Capsaicin enhances the drug sensitivity of cholangiocarcinoma through the inhibition of chemotherapeutic-induced autophagy. PLoS ONE, 2015, 10, e0121538.
  4. Wang, F.; Zhao, J.; Liu, D.; Zhao, T.; Lu, Z.; Zhu, L.; Cao, L.; Yang, J.; Jin, J.; Cai, Y. Capsaicin reactivates hMOF in gastric cancer cells and induces cell growth inhibition. Cancer Biol. Ther.2016, 17, 1117-1125.
  5. Huh, H. C.; Lee, S. Y.; Lee, S. K.; Park, N. H.; Han, I. S. Capsaicin induces apoptosis of cisplatin-resistant stomach cancer cells by causing degradation of cisplatin-inducible aurora-A protein. Nutr.Cancer.2011, 63, 1095–1103.
  6. Wang, H. M.; Chuang, S. M.; Su, Y. C.; Li, Y. H.; Chueh, P. J. Downregulation of tumor-associated NADH oxidase, tNOX (ENOX2), enhances capsaicin-induced inhibition of gastric cancer cell growth. Cell Biochem. Biophys. 2011, 61, 355–366.
  7. Anandakumar, P.; Kamaraj, S.; Jagan, S.; Ramakrishnan, G.; Asokkumar, S.; Naveenkumar, C.; Raghunandhakumar, S.; Devaki, T. Capsaicin inhibits benzo (a) pyrene-induced lung carcinogenesis in an in vivo mouse model. Inflamm. Res.2012, 61, 1169-1175.

Point 2: Row 59-there are not innumerable numbers!

Response: Changes made as told

Point 3: Row 63- the results could be quite disputable, let's hope they are not quiet

Response: had put appropriate word in the context

Point 4: Row 122- the table has not number nor legend!

Response: Have mentioned

Point 5: Row 145-148- please, reconsider the phrase!

Response: Statement reconsidered

Point 6: Rows 201-202 - the sentence has no predicate

Response: Statement has been made in a proper way

Point 7: Figure 1 has an uncomprehensible legend. what does it show? what is the connection with the text?

Response: Legend added to figure 1 and entire process has been described below

Point 8: Figure2 has an incomplete legend and has quite a poor design. please, reconsider design. also, there are no references, is it an original figure?

Response: Legend added to figure 2, the design is curated from the text written above figure 2 and thus its is original

Point 9: Figures 3 and 4 are original? if not, please add references!

Response: Figure 3 and 4 are diagrammatic representation of the text written above the figures 3 and 4 and thus they are original, but references added.

Point 10: Chapter 4.4.4.1. row 392- Factors influencing flavonoid concentration!

Response: Modified as told

Point 11: Table 3 has no references. is it an original one?

Response: We have added only 2 tables and references were added to each respective one, there is no table 3 in the manuscript

Reviewer 3 Report

Dear Authors,

It is my pleasure to review your article entitled "Antioxidant Activities Of Chilli: Biomedical Perspectives And Way Forward"; I went through the article and it is found to be scientifically sound and well-organized. However, I feel some editing in the manuscript can make it more attractive. I recommend to publish the manuscript in the journal after a major revision, because the article substantially contributes to the field of research and worthy of reading. My specific suggestions are the following;

1. Authors used statements like "important viral diseases", "important crop" etc. I suggest to remove these unwanted exaggerations.

2. The section 2.3 can be modified as "Traditional medicinal uses of Chilli" as the authors described more about the traditional applications of the plant.

3. Authors described about the traditional utility of the plant in Mayan, Latin american, and African medicines. What about Ayurveda? Is there any description on the use of Chilli in it. Include details.

4. under section 4.3 and 4.4, authors detailed about the major compounds of the plants. However, it would have been better if the authors can provide the quantitative profile of 6-8 major compounds.

5. The quality of figures 2, 3 and 4, especially their readability, needs to be improved (I think being a peer review version of the manuscript, the quality is reduced).

6. I suggest to include a consolidated figure on the entire biological activities of the bioactive compounds for an easier understanding.

Minor comments

Line 129: lutein are

Line 184: "as the" in bold

Line 534 (also in several other places): References are given as "[173, 174, 175]" If three or more continuous references are there, then authors may write it as [173-175]. Kindly see other manuscripts in the journal.

Several minor corrections including typographic mistakes and punctuation errors persists in the manuscript. I recommend authors to do some more time on proof reading before submitting revision.

Author Response

Point1. Authors used statements like "important viral diseases", "important crop" etc. I suggest to remove these unwanted exaggerations.

Response: Removed and replaced with suitable phrase “well known crops”

Point 2. The section 2.3 can be modified as "Traditional medicinal uses of Chilli" as the authors described more about the traditional applications of the plant.

Response: Changed as suggested

Point 3. Authors described about the traditional utility of the plant in Mayan, Latin american, and African medicines. What about Ayurveda? Is there any description on the use of Chilli in it. Include details.

Response: No description. Rather, as per Ayurvedic school of thought, chili is spicy in taste and hot by nature, it increases pitta dosha (Excess generation of heat in the body, acid reflux, gas, indigestion, Inflammation of the joints, nausea, diarrhea or constipation, anger & irritability, bad breath, body odor, excessive sweating). So needs to be used wisely and thus Ayurveda is apparently silent in this aspect. An extensive literature search in this aspects also revealed little mention of chilli in Ayurveda.

Point 4. under section 4.3 and 4.4, authors detailed about the major compounds of the plants. However, it would have been better if the authors can provide the quantitative profile of 6-8 major compounds.

Response: Quantitative profile of 6-8 major compounds added as asked to do

Point 5. The quality of figures 2, 3 and 4, especially their readability, needs to be improved (I think being a peer review version of the manuscript, the quality is reduced).

Response: Readibility has been improved substantially

Point 6. I suggest to include a consolidated figure on the entire biological activities of the bioactive compounds for an easier understanding.

Response: Added in the text as suggested

Point 7. Line 129: lutein are

Response: Done as told

Point 8. Line 184: "as the" in bold

Response: Done already

Point 9. Line 534 (also in several other places): References are given as "[173, 174, 175]" If three or more continuous references are there, then authors may write it as [173-175]. Kindly see other manuscripts in the journal.

Response: Done as suggested

Round 2

Reviewer 1 Report

The manuscript has been improved and can be published in present form, only I recommend to make the caption for figure 3 bold, like the rest of the figures.

Reviewer 2 Report

the present form of the work is satisfying. 

Reviewer 3 Report

It would have been better if the authors added the quantified data of bioactive compounds as a table.